# Evaluation of Online Inquiry Competencies of Chilean Elementary School Students: A Dataset

**Luz Chourio-Acevedo** [1,2,*] and **Roberto González-Ibañez** [1,2]

1   Departamento de Ingeniería Informática, Universidad de Santiago de Chile, Avenida Víctor Jara #3659, Santiago 9170124, Chile; roberto.gonzalez.i@usach.cl
2   InTeractiOn Lab, Universidad de Santiago de Chile, Avenida Víctor Jara #3659, Santiago 9170124, Chile
*   Correspondence: luz.chourio@usach.cl

**Abstract:** In the age of abundant digital content, children and adolescents face the challenge of developing new information literacy competencies, particularly those pertaining to online inquiry, in order to thrive academically and personally. This article addresses the challenge encountered by Chilean students in developing online inquiry competencies (OICs) essential for completing school assignments, particularly in natural science education. A diagnostic study was conducted with 279 elementary school students (from fourth to eighth grade) from four educational institutions in Chile, representing diverse socioeconomic backgrounds. An instrument aligned with the national curriculum, featuring questions related to natural sciences, was administered through a game named NEURONE-Trivia, which integrates a search engine and a logging component to record students' search behavior. The primary outcome of this study is a dataset comprising demographic information, self-perception, and information-seeking behaviors data collected during students' online search sessions for natural science research tasks. This dataset serves as a valuable resource for researchers, educators, and practitioners interested in investigating the interplay between demographic characteristics, self-perception, and information-seeking behaviors among elementary students within the context of OIC development. Furthermore, it enables further examination of students' search behaviors concerning source evaluation, information retrieval, and information utilization.

**Keywords:** information literacy; scholars; online search behaviors; online inquiry competencies





## 1. Summary

In the modern era, children and adolescents are growing up surrounded by technology. Specifically in Chile, findings from the Global Kids Online project indicate that the majority of respondents aged between 9 and 17 primarily utilize smartphones to access the internet (92%) [1]. This study also showed that the most prevalent online activities include entertainment and socializing, such as watching videos (91%), chatting (81%), using social networks (73%), playing games (69%), sharing multimedia content (59%), and communicating with family and friends (54%). Additionally, a vast majority of participants (84%) reported using the internet in the last month to complete school assignments. Despite the widespread use of technology, particularly the web, in both academic and non-academic contexts, there may be an insufficient development of information literacy competencies, particularly those essential for locating, evaluating, and effectively utilizing information to address certain academic challenges [2–6].

Information literacy (IL) comprises a set of competencies essential for learning [7], encompassing the skills, knowledge, and attitudes required to locate, evaluate, and effectively use information. It is also recognized as the ability to employ information effectively for problem-solving and decision-making [8–10]. It is noted that such competencies extend beyond printed materials to include digital content, data, images, and audio [11].

The COVID-19 pandemic accelerated the transition to online education, introducing new challenges for teachers, students, and their families. This shift exacerbated deficiencies in IL among both teachers and students [12], particularly evident when accessing online resources to fulfill various school assignments [13,14]. The vast quantity and diversity of online resources underscore the necessity for the proper development of online information competence (OIC).

An essential aspect of IL is the development of OICs. These competencies are pivotal for students to effectively locate, evaluate, and utilize information from the internet in problem-solving scenarios [15]. OIC encompasses a range of abilities, skills, and knowledge necessary to leverage the web as a research tool [16,17].

The information problem solving (IPS) process, involving the use of the internet (I), is delineated in the IPS-I model [18], which further elaborates on effective information-seeking strategies. This model integrates components from three prominent models: the Information Search Process (ISP) model [19], the Research Process Model [20], and the Big Six model [21]. The IPS-I model outlines five stages in the search process:

- Define Information Problem: This initial phase involves clearly defining the problem, formulating the main question and sub-questions, considering requirements, and activating prior knowledge. A well-defined problem is crucial for finding adequate solutions and answers;
- Search Information: Students select a search strategy, specify search terms, and evaluate the websites from the search results. Common strategies include using a search engine, typing a URL directly, or browsing through links. Effective search terms are specific and informed by the students' prior knowledge, facilitating a more targeted search;
- Scan Information: After identifying potential sources, students scan the information to assess its usefulness. This involves quickly reviewing the content and integrating it with prior knowledge or other found information. Useful information is bookmarked or copied for later use;
- Process Information: This phase involves deep processing to achieve a thorough understanding and integration of the information with prior knowledge. Students analyze, select, and structure information, using criteria to judge its usefulness and quality;
- Organize and Present Information: In the final synthesis phase, students combine all gathered information to solve the information problem. They create a product, such as a presentation, poster, or written essay, ensuring the problem is well-formulated and the information is well-organized and elaborated upon.

Research on IL in the internet age presents several challenges and, simultaneously, offers new opportunities to study how students approach information problems using online resources [22,23]. In this context, examining the online search behaviors, cognitive and affective processes, and motivational factors of students engaged in online inquiry provides novel perspectives for understanding strategies, problems, and patterns, among other aspects of online inquiry [24]. These insights can inform the development of models and support systems aimed at developing students' competencies to locate, evaluate, and use information effectively across various activities.

Taking a closer examination of the Chilean educational landscape, we initiated a cross-sectional study to assess elementary school students' OIC within a near-realistic context. Our diagnostic framework encompasses a spectrum of information challenges within the natural sciences, stratified by grade level (i.e., fourth, sixth, and eighth grades), question category (i.e., closed questions, descriptive, causal explanation, verification, prediction, difficulty (i.e., easy, intermediate, hard), and generalization [25]), and challenge type (i.e.,

open answer and source evaluation). To engage students in the diagnostic process while maintaining control over the information resources accessed during search sessions, we utilized the Trivia game [15], powered by the NEURONE ecosystem [26]. Consequently, both the challenges and the repository of information resources (e.g., web pages, images, and videos) were loaded into the Trivia platform. This facilitated an experiential learning environment for students, simulating search experiences akin to those provided by contemporary search engines such as Google and Bing.

In 2022, the diagnostic was administered to 169 students from three public schools and 110 students from a private educational institution within the metropolitan area of Chile. This article describes the resulting dataset, encompassing demographic, self-perception, and behavioral data derived from 1056 search sessions.

The subsequent sections of this article are structured as follows: Section 2 elucidates the contents of the dataset. Following this, Section 3 delineates the methodology employed, along with the resources and instruments utilized in the data collection process. Lastly, Section 4 presents the conclusions drawn and potential directions for future research.

## 2. Data Description

The following three sections provide a detailed description of the dataset. Firstly, we introduce the files comprising the dataset. Secondly, we describe the various features within each file. Finally, we offer an overview of the distribution of the data.

### 2.1. Files

The dataset is structured into five comma-separated values (.csv) files. A comprehensive list of these files, along with concise descriptions for each, is presented in Table 1.

**Table 1.** List of files and their description.

| Filename | Description |
|---|---|
| challenges.csv | Contains the full list of challenges used in the diagnostic. It includes grade level, question category, and type. |
| resources.csv | Includes the list of resources (i.e., webpages, videos, and PDF documents) associated with each challenge, along with information regarding their relevance to it. |
| answerTypeResponses.csv | Comprises the list of actual answers that students provided to solve the challenges. |
| sourceTypeResponses.csv | Comprises the list of sources that students found to be relevant to tackle the challenges along with justifications for their selection. |
| queries.csv | Corresponds to the list of all the search queries that students issued while addressing the challenges. |

### 2.2. Data Fields

Each file listed in Table 1 contains data recorded in a tabular format during the diagnostic process. Rows represent individual entries, while columns (separated by commas) correspond to fields. The file containing the entries for challenges (challenges.csv) comprises 10 fields, which are detailed in Table 2.

The second file (resources.csv) contains the list of resources associated with each challenge. Each row provides information about the challenge to which the resource is linked, its relevance to the challenge, its type, and the URL from which the resource was obtained. An overview of this file's structure is presented in Table 3.

Table 4 describes the list of common fields found in answerTypeResponses.csv and sourceTypeResponses.csv. These include aggregated behavioral data collected during search sessions (such as the number of visited pages and queries) as well as questionnaire data (demographic information, pre- and post-challenge responses). Subsequently, Tables 5 and 6 outline specific fields within these files, containing students' responses to both types of challenges (text answers and source selection).

**Table 2.** Fields for challenges (challenges.csv).

| Field Name | Type | Description |
| --- | --- | --- |
| challegeId | Identifier | Unique identifier for the challenge |
| gradeLevel | Categorical | Grade level in the Chilean education system |
| challengeCode | Categorical | Descriptive code that combines representative characters of the constituent elements of a challenge, namely, Level: 4B, 6B or 8B. Question Type: Descriptive or Closed Question (D), Causal Explanation or Verification (E), Prediction or Generalization (P) Answer Type: Text (T) and Justification (J) (e.g., 8BDT1, 4BEJ2, 4BPT3) |
| challengeType | Categorical | Challenge type in Trivia: text answer (A), source selection with justification (S) |
| challengeCategory | Categorical | Descriptive, Closed Question, Causal Explanation, Verification, Prediction or Generalization [25] |
| challengeDifficulty | Categorical | Challenge difficulty rated by expert judgment: low (L), medium (M) or high (H). |
| challengeQuestion | Categorical | The verbatim of the challenge, expressed as a question or requirement. |
| challengeExpectedAnswer | Categorical | Expected answer according to the associated relevant source indicated in resources.csv. |
| challengeQuestionEn | Categorical | Translation of challengeQuestion to American English. |
| challengeExpectedAnswerEn | Categorical | Translation of challengeExpectedAnswer to American English. |

**Table 3.** Fields for resources (resources.csv).

| Field Name | Type | Description |
| --- | --- | --- |
| resourceCode | Identifier | Descriptive unique code that combines representative characters of the constituent elements of a resource, namely, Educational level: 4B, 6B, 8B, 2M, 4M Question number: 1, 2, 3, 4, 5, . . . Resource Type: web page (P), video (V), or PDF book (L), Source type: Reliable(C), Alternative(A). Resource number: Sequential number (1, 2, 3, . . . 4) (e.g., 4B2LC1, 4B2PA2, 4B2PA3). |
| challengeId | Identifier | The ID of the challenge to which the resource is associated. |
| resourceRelevance | Binary | It indicates whether the resource for solving a challenge comes from a reliable source (1: yes, 0: no) |
| resourceType | Categorical | It indicates the type of resource: web page (HTML), video, or PDF. |
| URL | Categorical | Uniform Resource Locator (URL) from where the resource was obtained. |

**Table 4.** Common fields found in answerTypeResponses.csv and sourceTypeResponses.csv.

| Field Name | Type | Description |
| --- | --- | --- |
| userId | Identifier | Student's unique identifier. |
| school | Categorical | Educational institution to which the student belongs. |
| Sex | Categorical | Sex declared by the student (M: male, F: female). |
| Age | Numeric | Student's age. |
| gradeLevel | Categorical | Students' grade according to the Chilean educational system. |

**Table 4.** *Cont.*

| Field Name | Type | Description |
|---|---|---|
| classSection | Categorical | Class section. Institution may encompass one or multiple sections for a grade level. This is expressed by alphabetical labels (e.g., A, B, C). |
| challengeId | Identifier | The identifier of the challenge to which the student responded. |
| prePriorKnowledge | Categorical | Indicates self-reported prior knowledge about the challenge as declared before searching for information ("I know a lot about the topic of the challenge I am going to do") *. |
| preMotivation | Categorical | Indicates how motivated the student felt before addressing the challenge ("I am motivated by the challenge I am going to solve.") *. |
| prePerceivedDifficulty | Categorical | Indicates the perceived difficulty level of the challenge by the student just before beginning to work on it ("The challenge I am going to do is difficult") *. |
| preMood | Categorical | Indicates the degree of contentment, joyfulness, liveliness, or happiness experienced by the student just before commencing work on the challenge ("How happy, joyful, or uplifted do you feel now?") *. |
| challengeQuestion | Categorical | The verbatim of the challenge, expressed as a question or requirement. |
| postPerceivedPerformance | Categorical | Indicates the level of confidence the student had in the correctness of the provided answer, if any ("The results I achieved in the challenge were good") *. |
| postMotivation | Categorical | Indicates how motivating the challenge was for the student during its completion ("The challenge I just completed was motivating") *. |
| postDifficulty | Categorical | Indicates the perceived difficulty level of the challenge by the student immediately after its completion or running out of time ("The challenge was difficult") *. |
| postMood | Categorical | Indicates the degree of contentment, joyfulness, liveliness, or happiness experienced by the student immediately after completing the challenge or upon running out of time ("How happy, joyful, or lively do you feel now?") **. |
| firstQueryTime | Numeric | Time elapsed from the initiation of the challenge until the first keystroke is entered in the query box (expressed in seconds). |
| writingQueriesTime | Numeric | Total time (expressed in seconds) spent formulating queries. When students perform a copy/paste action in the query box, the resulting value is zero. |
| qMod | Numeric | Number of query reformulations per challenge [27]. |
| countQueries | Numeric | Total number of queries per challenge [28]. |
| stayPagesRelv | Numeric | Total time (expressed in seconds) spent on relevant pages. |
| stayPagesNotRelv | Numeric | Total time (expressed in seconds) spent in non-relevant pages. |
| stayPagesSerp | Numeric | Total time (expressed in seconds) spent on search engine results pages (SERPs). |
| clicRelv | Numeric | Number of clics on relevant documents. |
| clicNotRelv | Numeric | Number of clics on non-relevant documents. |
| clicSerp | Numeric | Number of clics on SERPs. |
| mouseMovRelv | Numeric | Number of mouse movements in relevant documents. |

**Table 4.** *Cont.*

| Field Name | Type | Description |
| --- | --- | --- |
| mouseMovNotRelv | Numeric | Number of mouse movements in non-relevant documents. |
| mouseMovSerp | Numeric | Number of mouse movements in SERPs. |
| scrollRelv | Numeric | Number of scrolling actions within relevant pages. |
| scrollNotRelv | Numeric | Number of scrolling actions within non-relevant pages. |
| scrollSerp | Numeric | Number of scrolling actions within SERPs. |
| totalSearchTime | Numeric | Time spent solving the task, expressed in seconds, with a maximum limit of 300 s. |
| totalCover | Numeric | Number of informational resources accessed throughout the challenge [28]. |
| usfCover | Numeric | Number of informational resources accessed throughout the challenge with dwell time equal to or greater than 30 s [28]. |
| docRelvVist | Numeric | Number of relevant informational resources accessed throughout the challenge. |

* Likert scale from 1 to 6, where 1 represents "Strongly disagree" and 6 "Strongly agree". ** Likert scale from 1 to 6, where 1 represents "Little or nothing" and 6 "A lot".

**Table 5.** Specific fields for students' responses for text-answer-type challenges (answerTypeResponses.csv).

| Field Name | Type | Description |
| --- | --- | --- |
| expectedAnswer | Categorical | Expected answer according to the associated relevant source indicated in resources.csv. |
| studentAnswer | Categorical | The answer provided by the student. |

**Table 6.** Specific fields for students' responses for source-selection-type challenges (sourceTypeResponses.csv).

| Field Name | Type | Description |
| --- | --- | --- |
| relevantPage | Categorical | Web address of the relevant page |
| Justify | Categorical | Justification of page selection |
| selectedPage | Categorical | Page selected by the user |

Table 7 presents the list of search queries formulated by students during their engagement with the challenges.

**Table 7.** Fields for students' queries (queries.csv).

| Field Name | Type | Description |
| --- | --- | --- |
| userId | Identifier | Student's unique identifier. |
| challengeId | Identifier | The identifier of the challenge to which the student responded. |
| Query | Categorical | Query formulated by the user. |

*2.3. Data Distribution*

To better contextualize the dataset, this section provides descriptive data of the educational institutions and participants.

As depicted in Table 8, four schools from various districts of Santiago, the capital of Chile, participated in the diagnostic assessment. In terms of sector, three were public, and only one was private. To provide a better understanding of the socioeconomic disparities among these schools, the School Vulnerability Index (IVE—Índice de Vulnerabilidad Escolar) is also included. This index, calculated by the National Board of School Aid and Scholarships (JUNAEB—Junta Nacional de Auxilio Escolar y Becas), is used to identify and classify educational institutions based on the socioeconomic vulnerability levels of their

students. The IVE assesses vulnerability by considering various risk factors such as health and poverty, thereby informing government policies. The higher the index, the greater the vulnerability. This approach illustrates inequalities within the school environment, linking them to structural social conflicts and socioeconomic status [29].

**Table 8.** Demographic data of the participating institutions in the diagnostic of OIC.

| Institution | Commune | IVE * | Sector |
|---|---|---|---|
| Institution 1 | Providencia | 58% ** | Private |
| Institution 2 | Huechuraba | 91% | Public |
| Institution 3 | Quinta Normal | 83% | Public |
| Institution 4 | La Reina | 70% | Public |

* School Vulnerability Index (JUNAEB) source: https://www.junaeb.cl/ive-2022 (accessed on 10 November 2023).
** The facility is not included in the IVE list, so the commune's IVE was used as a reference.

The diagnostic assessment was conducted in accordance with the classes agreed upon with each school, ranging from fourth to eighth grade in elementary school education. In total, 347 students aged between 9 and 15 participated in the diagnostic assessment, with a balanced distribution between girls and boys (a difference of 1.8%), as illustrated in Table 9.

**Table 9.** Demographic data of the students who participated in the diagnostic study.

| | |
|---|---|
| Grades | 4th to 8th grade of elementary school |
| Ages | Between 9 and 15 years old |
| Number of participants | 347 |
| Consents received | 279 |
| Girls | 49.1% |
| Boys | 50.9% |

An important ethical consideration in this study was the participation of underage students, necessitating both their assent and the consent of their guardians. Students were provided with a printed informed consent form, which necessitated the signature of their guardian upon return. Throughout the diagnostic phase, all students in the class participated uniformly. Subsequently, the collected data were filtered based on the consent obtained from their respective guardians.

Lastly, Figure 1 illustrates the distribution of participants by school and grade level. Most institutions had only one section per class, except for Institution 1, which had four sections of the same level (fourth grade).

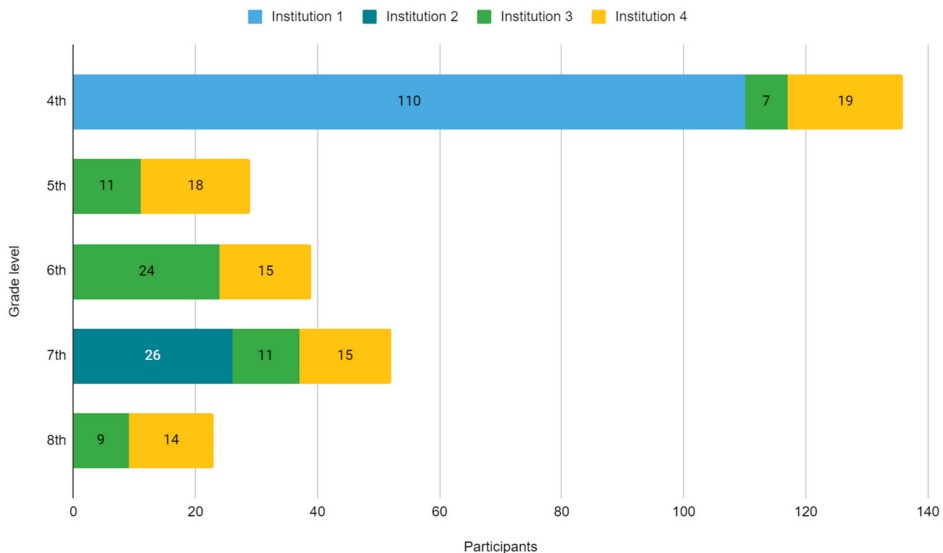

**Figure 1.** Participants by institution and grade.

## 3. Methods

In the following section, we provide background information on the research objective, study design, resources, instruments, tools, and data collection procedures.

### 3.1. Research Objective

The objective of this study was to assess the developmental level of OIC among elementary school students in Chile within the domain of natural sciences.

### 3.2. Study Design

The diagnostic study was designed as an interactive experience, with each student tasked with solving four challenges in the field of natural sciences and completing questionnaires. The presentation order of challenges followed a rotational design, which was automatically assigned during the creation of student accounts within Trivia. Students were given a maximum of five minutes (300 s) to complete each challenge, totaling 20 min. Additionally, they were provided with time for receiving instructions via a video tutorial, engaging in practice challenges, and responding to pre- and post-challenge questionnaires assessing prior knowledge, affective states, motivation, perceived difficulty, and confidence in task performance using a 6-point Likert scale. The diagnostic was tailored to be completed within 60 min or less, aiming to fit within a regular class period while also minimizing student fatigue and boredom.

The study presented two types of challenges. In the first type (A), students were required to locate a specific answer within the document collection and provide it using a text form entry. In the second type (S), students had to bookmark a reliable source containing the answer to the challenge and justify their selection. Examples of these two types of challenges are listed in Table 10.

**Table 10.** Examples of challenges indicating type (A: Answer, S: Source selection), category (D: Description, E: Causal explanation), difficulty (E: Easy, I: Intermediate), and elementary school grade level.

| Challenge | Type | Category | Difficulty | Grade Level |
|---|---|---|---|---|
| Write the answer to the following question: What name receives the natural layer located in the upper atmosphere that covers the planet Earth protecting it from external UV radiation? For this question, locate a reliable source that provides the answer, and briefly explain why you consider it reliable. | A | D | E | 4th |
| Why does sports help keep your heart healthy? | S | C | I | 6th |

The challenges were tailored to meet the requirements of the Chilean elementary and middle school curricula, specifically addressing natural sciences for grades four, six, and eight. The application of these challenges was as follows: those designed for sixth grade were also implemented for fifth graders, and similarly, those intended for eighth grade were extended to seventh graders.

Each challenge was expressed in Spanish and associated with at least three informational resources (i.e., webpages, videos, and PDF documents). One of the resources served as a reliable source containing the answer to the challenge, while the others acted as unreliable alternative sources containing either the wrong answer or part of it. The selection and curation of the resources were carried out by four researchers from our team. Subsequently, the list of challenges and their respective resources underwent review by two education experts, aiming to ensure clarity, relevance, and adequacy for the target grade levels of the diagnostic [19]. The adjustments proposed by the experts contributed

to improving the presentation of the questions, thereby ensuring that participants could effectively understand the information-seeking tasks assigned to them.

### 3.3. Technological Tool

The diagnostic was conducted utilizing the Trivia game platform [15] to offer students an engaging experience. The platform incorporates the NEURONE search engine, which allows students to search for information within a curated collection of resources using a user interface similar to popular search engines (e.g., Google, Bing).

To collect the data included in this dataset, the NEURONE logger [26] was activated within Trivia to capture various types of data, such as keyboard entries, bookmarks, visited pages, search queries, mouse movements, scrolling, and time spent on pages, as discussed in the previous section.

As depicted in Figure 2, the user interface, through which challenges were presented to students, was inspired by "A Google a day". The bottom area of the interface presents students with the challenge, remaining time, and the answer form. The central area provides direct access to the NEURONE search engine, and it is also where informational resources are deployed. Finally, the header contains a home button, access to help, and a button to end the session.

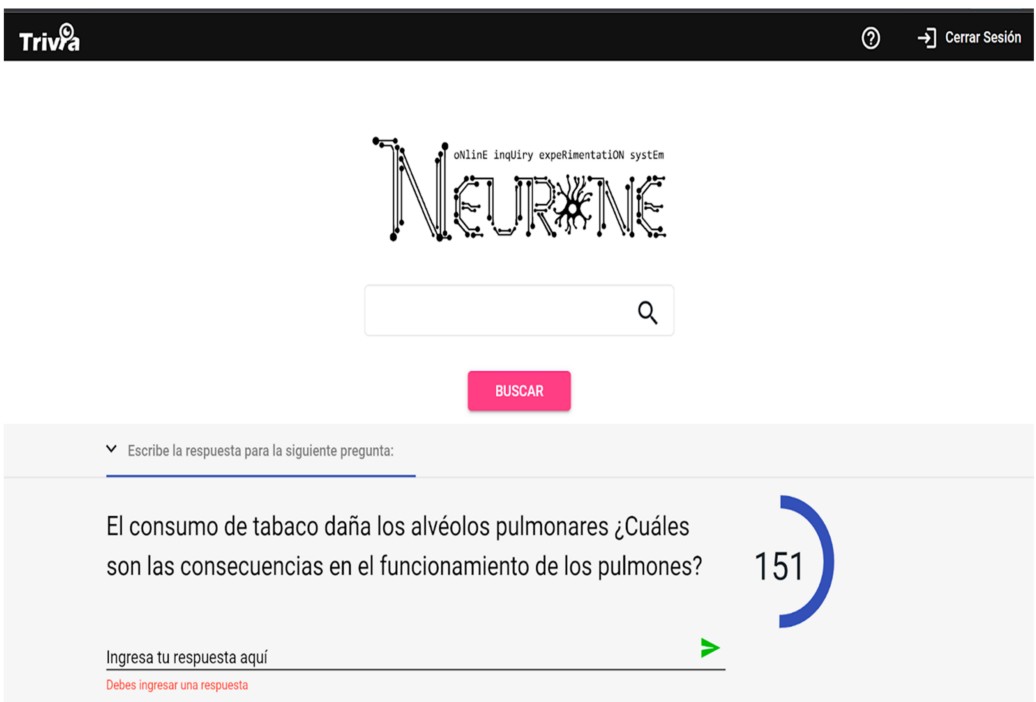

**Figure 2.** Challenge (answer type) in Trivia.

### 3.4. Information Search Metrics

Information search metrics were categorized into three groups: demographic data, self-reported data, and behavioral data. Each group of variables plays an important role in measuring the OIC. The specific variables are described in detail in Section 2.2.

Demographic Data:

These variables provide context about the students' background, which is essential for understanding the diversity and potential differences in information-seeking behaviors:

- school, sex, age, gradeLevel, classSection: These variables help to identify patterns and correlations between students' demographic characteristics and their OIC.

Self-Reported Data:

Collected through pre- and post-challenge questionnaires, these variables reflect the students' subjective experiences and attitudes during the search process. They are crucial for understanding motivational and affective factors influencing OIC [30]:

- Pre-Challenge (priorKnowledge, preMotivation, prePerceivedDifficulty, preMood): These variables help gauge the students' initial readiness and attitude towards the task, which can impact their search effectiveness and persistence;
- Post-Challenge (postPerceivedPerformance, postMotivation, postDifficulty, postMood): These variables provide insights into the students' self-assessment and emotional response after completing the task, indicating how the experience might influence their future information-seeking behaviors.

Behavioral Data:

These variables are automatically captured during the search process and provide objective measures of the students' interactions with online resources. They are categorized according to the IPS-I model stages:

- Define Information Problem:
  - firstQueryTime: Measures how quickly a student begins the search, indicating their initial understanding and confidence in defining the problem [31];
- Search Information:
  - writingQueriesTime, qMod, countQueries: These metrics assess the student's ability to formulate effective search queries and adapt their search strategy as needed [18,28,32];
  - queries: Specific terms used by students, reflecting their ability to generate relevant keywords;
- Scan Information:
  - stayPagesRelv, stayPagesNotRelv, stayPagesSerp: These metrics indicate the student's ability to identify useful information quickly;
  - clicRelv, clicNotRelv, clicSerp, mouseMovRelv, mouseMovNotRelv,
  - mouseMovSerp, scrollRelv, scrollNotRelv, scrollSerp: These actions reflect how students interact with search results and web pages, showing their scanning efficiency and focus [7,18,32];
- Process Information:
  - totalSearchTime, totalCover: These metrics provide a holistic view of the student's thoroughness in gathering information;
  - usfCover, docRelvVist: These indicate the student's ability to identify and focus on high-quality information sources [18,28];
- Organize and Present Information:
  - studentAnswer, justification, selectedPage: These metrics evaluate how well students synthesize and present the information they have gathered, demonstrating their ability to effectively use the information to solve the problem [18].

By examining these variables, comprehensive information on the development of OIC among elementary students can be obtained by assessing their ability to effectively define problems and search, evaluate, process, and present information.

### 3.5. Data Collection

The activities necessary for data collection were divided into three stages:

- Before conducting the diagnostic at each institution, the research team assessed the technological infrastructure and spatial conditions necessary for the study. Subsequently, user accounts were created in Trivia to encompass all students in the participating classes of the school;
- On the day of the diagnostic, students were assigned one of the user accounts previously created for them. Subsequently, they received guidance on navigating the system

through a video tutorial and a presentation led by one of the researchers. Following this, students were given the opportunity to complete two practice challenges (one for each challenge type) to become acquainted with the system. Throughout the sessions, connectivity and other technical issues were monitored and addressed as feasible. The researchers conducting the study meticulously documented all encountered issues. Following the completion of the class diagnostic, open-ended questions were posed to the students to elicit qualitative feedback on their experience;

- After completing a class diagnostic, the consistency between the attendance list and the signed consents was verified. Subsequently, the data were filtered, organized, and aggregated based on the consents received for the creation of the presented dataset. It is noteworthy that all identifiable data collected during the study were removed from the dataset.

## 4. User Notes

This article introduces a novel dataset encompassing demographic information, self-perception, and information-seeking behaviors data collected during an OIC diagnostic. The data were gathered from 279 Chilean elementary school students attending four elementary schools with diverse socioeconomic backgrounds.

The dataset is designed to facilitate researchers' exploration of the interplay between cognitive, affective, and behavioral data among elementary school students as they engage in natural sciences tasks. Additionally, further analyses can be conducted using this dataset to delineate students' search behaviors pertinent to OIC, particularly those concerning source evaluation, information retrieval, and information use.

This dataset offers valuable insights with practical implications for educational practices. It can guide curriculum developers in integrating OIC into educational frameworks. By analyzing the data, educators can tailor curriculum content to enhance information literacy skills, incorporating activities that promote critical thinking, source evaluation, and effective information retrieval, thereby better preparing students to navigate the internet.

Analysis of this data can also inform the allocation of resources and support systems in schools. Schools can use the data to identify areas needing additional training or resources to help students develop online inquiry skills. This might involve providing access to online databases, offering workshops on information search strategies, or creating collaborative spaces for research projects. Aligning school resources with the development of students' OIC can foster a conducive environment to academic growth and information literacy.

Moreover, the dataset can shed light on the impact of online inquiry skills on students' well-being. By identifying patterns in students' affective states, motivation levels, and perceived difficulty during research sessions, educators can implement strategies to support students' mental and emotional health. Promoting a positive learning environment, offering guidance on managing search-related stress, and fostering a sense of achievement in information retrieval tasks can contribute to students' overall well-being and positively influence their lifelong learning.

**Author Contributions:** Conceptualization, L.C.-A. and R.G.-I.; methodology, L.C.-A. and R.G.-I.; validation, L.C.-A.; formal analysis, L.C.-A. and R.G.-I.; investigation, L.C.-A. and R.G.-I.; resources, L.C.-A.; data curation, L.C.-A.; writing—original draft preparation, L.C.-A.; writing—review and editing, L.C.-A. and R.G.-I.; supervision, R.G.-I. All authors have read and agreed to the published version of the manuscript.

**Funding:** This research was funded by the TUTELAGE project, FONDECYT Regular [grant no. 1201610], funded by ANID and the Vicerrectoría de Postgrado de la Universidad de Santiago de Chile, BECA DE EXCELENCIA PARA EXTRANJEROS (2019).

**Institutional Review Board Statement:** The study was approved by the Ethics Committee of Universidad de Santiago de Chile (Ethical Report No. 221/2020).

**Informed Consent Statement:** Informed consent was obtained from all subjects involved in the study.

**Data Availability Statement:** The dataset is available in the Mendeley Data repository. Doi: 10.17632/dwkh4yfm7f.4. The data are licensed under a Creative Commons Attribution 4.0 International (CC BY 4.0), which permits use, distribution, and reproduction in any medium, provided the original work is properly cited.

**Conflicts of Interest:** The authors declare no conflicts of interest.

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
