# Peer review of "Evaluation of Online Inquiry Competencies of Chilean Elementary School Students: A Dataset"

_data, 2024_

Round 1

Reviewer 1 Report

Comments and Suggestions for Authors

The article describes the general research concept and organization of the database. This is interesting from the point of view of the research tools used. However, the research procedure is incomplete, there are no clear results and no indications for drawing conclusions.

Author Response

The article describes the general research concept and organization of the database. This is interesting from the point of view of the research tools used. However, the research procedure is incomplete, there are no clear results and no indications for drawing conclusions.

Given that our article is a data descriptor, the author guidelines indicate that it should include the following sections: “Summary, Data Description, Methods, User Notes.” Consequently, the sections “results” and “conclusions” are not present, as we are only describing a new dataset and do not present research results.

Reviewer 2 Report

Comments and Suggestions for Authors

Title:

Evaluation of Online Inquiry Competences of Chilean Elementary School Students: A Dataset

The reviewer’s comments

1.     The article shows sufficient reflections and discussions about the research topic

on evaluation of online inquiry competences of Chilean elementary school students

2.     The abstract will be revised to include details pertaining to various aspects such as data collection, participants, methodologies employed, and the instruments utilised

3.     The theoretical perspectives framing the article are not clear.

Please strengthen this, and elaborate on that.

4.     The authors need to expand and elaborate on the User Notes. What does that mean in practice, for the curriculum, for the organization of the campus, and for the well-being of the students? Examples are welcome.

5. I am not saying the article should be rejected now, but it has to embark on major changes before being included in this publication. Revisions or explanations are all made according to the suggestions of the reviewer. Requires Major Revision.

Comments on the Quality of English Language

Title:

Evaluation of Online Inquiry Competences of Chilean Elementary School Students: A Dataset

The reviewer’s comments

1.     The article shows sufficient reflections and discussions about the research topic

on evaluation of online inquiry competences of Chilean elementary school students

2.     The abstract will be revised to include details pertaining to various aspects such as data collection, participants, methodologies employed, and the instruments utilised

3.     The theoretical perspectives framing the article are not clear.

Please strengthen this, and elaborate on that.

4.     The authors need to expand and elaborate on the User Notes. What does that mean in practice, for the curriculum, for the organization of the campus, and for the well-being of the students? Examples are welcome.

5. I am not saying the article should be rejected now, but it has to embark on major changes before being included in this publication. Revisions or explanations are all made according to the suggestions of the reviewer. Requires Major Revision.

Author Response

  • The abstract will be revised to include details pertaining to various aspects such as data collection, participants, methodologies employed, and the instruments utilised.

The abstract has been revised to include detailed information on various aspects such as data collection, participants, methodologies employed, and the instruments utilized. All suggestions have been incorporated into the new version of the abstract.

  • The theoretical perspectives framing the article are not clear. Please strengthen this, and elaborate on that.

The theoretical perspectives framing the article have been clarified and strengthened. In Section 1, 'Summary,' we explain Online Inquiry Competences (OIC) as a fundamental aspect of contemporary Information Literacy, along with the stages of the IPS-I model, which underpins this work. Furthermore, in Section 3.4, 'Information Search Metrics,' we detail the metrics captured for each stage of the IPS-I.

  • 4. The authors need to expand and elaborate on the User Notes. What does that mean in practice, for the curriculum, for the organization of the campus, and for the well-being of the students? Examples are welcome.

Section 4, has been expanded to address practical implications for curriculum development. We included examples of how the data can guide curriculum adjustments, resource allocation, and support strategies to enhance students' online inquiry skills and overall well-being.

Reviewer 3 Report

Comments and Suggestions for Authors

This article need to add relative works or literature ( theoretical based) related to 

(1) online inquiry competences (OIC)?

(2)the constructs or dimensions of OIC and how to measure OIC?

(3)what is the relation between dataset (variable  or field) iand OIC measurement ?

Comments on the Quality of English Language

Minor editing of English language required.

Author Response

  • (1) online inquiry competences (OIC)?

The definition of Online Inquiry Competences (OIC) is included in Section 1, 'Summary.' This section also outlines the stages of the IPS-I model, which helps in detailing the constructs to be measured.

  • (2) the constructs or dimensions of OIC and how to measure OIC?

Section 3.4 presents the constructs of OIC and how to measure them. It details the metrics captured by the Neurone-TRIVIA system based on the stages of the IPS-I model.

  • (3) what is the relation between dataset (variable  or field) and OIC measurement ?

Sections 1 and 3.4 explain the relationship between the dataset variables and OIC measurement. Section 1 outlines the IPS-I model stages used to measure OIC, and Section 3.4 describes how the presented metrics evaluate students' OIC levels.

Round 2

Reviewer 3 Report

Comments and Suggestions for Authors

Accept in present content.